# Fabric-based lamina emergent MXene-based electrode for electrophysiological monitoring

Sanghyun Lee[1,7], Dong Hae Ho[2,7], Janghwan Jekal [3,7], Soo Young Cho [1], Young Jin Choi [1], Saehyuck Oh [3], Yoon Young Choi[4], Taeyoon Lee [5,6], Kyung-In Jang [3] ✉ & Jeong Ho Cho [1] ✉

Commercial wearable biosignal sensing technologies encounter challenges associated with irritation or discomfort caused by unwanted objects in direct contact with the skin, which can discourage the widespread adoption of wearable devices. To address this issue, we propose a fabric-based lamina emergent MXene-based electrode, a lightweight and flexible shape-morphing wearable bioelectrode. This work offers an innovative approach to biosignal sensing by harnessing the high electrical conductivity and low skin-to-electrode contact impedance of MXene-based dry electrodes. Its design, inspired by Nesler's pneumatic interference actuator, ensures stable skin-to-electrode contact, enabling robust biosignal detection in diverse situations. Extensive research is conducted on key design parameters, such as the width and number of multiple semicircular legs, the radius of the anchoring frame, and pneumatic pressure, to accommodate a wide range of applications. Furthermore, a real-time wireless electrophysiological monitoring system has been developed, with a signal-to-noise ratio and accuracy comparable to those of commercial bioelectrodes. This work excels in recognizing various hand gestures through a convolutional neural network, ultimately introducing a shape-morphing electrode that provides reliable, high-performance biosignal sensing for dynamic users.

Wearable technology is substantially gaining importance in the healthcare sector owing to various societal concerns, such as aging populations and the emergence of infectious diseases. Preventive medicine has the potential to reduce social costs significantly by enabling early interventions against acute diseases[1]. However, the effective implementation of preventive medicine faces several challenges, including difficulties in miniaturizing biodata acquisition systems, ensuring stable skin-to-bioelectrode contact, allowing continuous wear, and facilitating data analysis[2]. To address these challenges, wearable healthcare systems have emerged as prominent candidates that can seamlessly collect and track user biodata[3,4].

Among the wearable healthcare systems, tattoo-based[5–7], and textile-based[8,9] wearables have been extensively studied because of their potential for daily analysis of physiological diseases, such as

[1]Department of Chemical and Biomolecular Engineering, Yonsei University, Seoul, Republic of Korea. [2]Department of Energy Science and Engineering, Daegu Gyeongbuk Institute of Science and Technology (DGIST), Daegu, Republic of Korea. [3]Department of Robotics and Mechatronics Engineering, Daegu Gyeongbuk Institute of Science and Technology (DGIST), Daegu, Republic of Korea. [4]Department of Mechanical Science and Engineering, University of Illinois at Urbana–Champaign, Urbana, IL, USA. [5]School of Electrical and Electronic Engineering, Yonsei University, Seoul, Republic of Korea. [6]Department of Bio and Brain Engineering, Korea Institute of Science and Technology (KIST), Seoul, Republic of Korea. [7]These authors contributed equally: Sanghyun Lee, Dong Hae Ho, Janghwan Jekal. ✉e-mail: kijang@dgist.ac.kr; jhcho94@yonsei.ac.kr

cardiovascular diseases[10], muscle disorders[11], Parkinson's disease[12], and Alzheimer's disease[13]. Recent advancements in stretchable and flexible dry electrodes, which comprise one-dimensional (1D) or two-dimensional (2D) conductive materials, have significantly improved their sensitivity, particularly for body suit-type applications[14,15]. Despite their advantages, wearable devices have persistently caused issues of discomfort due to direct contact with the skin. Tokihiko et al. highlighted the importance of comfort-of-wear as a critical parameter for the future of wearables, similar to selecting comfortable clothing[16]. This term describes the ability of a device to seamlessly integrate into a user's daily life without causing any disruption or sense of irregularity because of its physical presence. Although the size of current rigid wearable devices, such as wristwatches or rings, has been reduced to enhance comfort, the discomfort associated with wearing such devices has failed to be eliminated[16]. Consequently, individuals sensitive to discomfort caused by irregular objects are discouraged from using these devices, which in turn deters the adoption of wearable healthcare devices and limits the effectiveness of preventive medicine efforts.

To tackle this issue, we present a fabric-based lamina emergent MXene-based electrode (FLEXER) as a wearable biosensing electrode. This shape-morphing bioelectrode utilizes a laminar emergent mechanism crafted from thermoplastic polyurethane (TPU)-laminated cotton fabric. The FLEXER has a symmetric whirlpool-shaped configuration that seamlessly transforms from a flat state to a scissor jack-like shape when pneumatic pressure is applied. The expanded air pouch self-intersects, forming millimeter-sized joints that can be individually folded and unfolded, enabling conformal and stable electric contact with the skin even during motion. The contact force of the FLEXER can be adjusted by controlling the air pressure and geometric characteristics of the joints. Additionally, the primary electrode material of the FLEXER is MXene, a 2D material comprising transition metal and carbon with excellent electrical conductivity and mechanical durability[17]. To further enhance the mechanical properties[18] and oxidation resistance[19,20] of MXene, it was supplemented with cellulose nanofiber (CNF)[21] and polycarboxylate ether (PCE)[22]. This innovative FLEXER technology can seamlessly integrate into everyday clothing, allowing for noninvasive monitoring of electrocardiogram (ECG) and electromyogram (EMG) signals. By combining the advantages of pneumatic lamina emergent systems and flexible MXene electrodes, we were able to develop a concealable, fast, safe, and reliable biosensing electrode that revolutionizes the field of wearable health monitoring.

## Results

### Concept of FLEXER as a wearable device

The FLEXER-based wearable healthcare system differs from conventional wearable healthcare systems in that it has a concealable design owing to its shape-morphing capability (Fig. 1a (left)). Unlike other wearable systems that use bioelectrodes requiring constant direct skin contact, the FLEXER eliminates the constant need for physical contact. Traditional bioelectrodes, particularly dry ones, require pressure application, which is usually accomplished with biomedical adhesives or pressure bandages[23]. However, the unique design of the FLEXER allows it to be activated or deactivated as required (see Fig. 1a (right)),

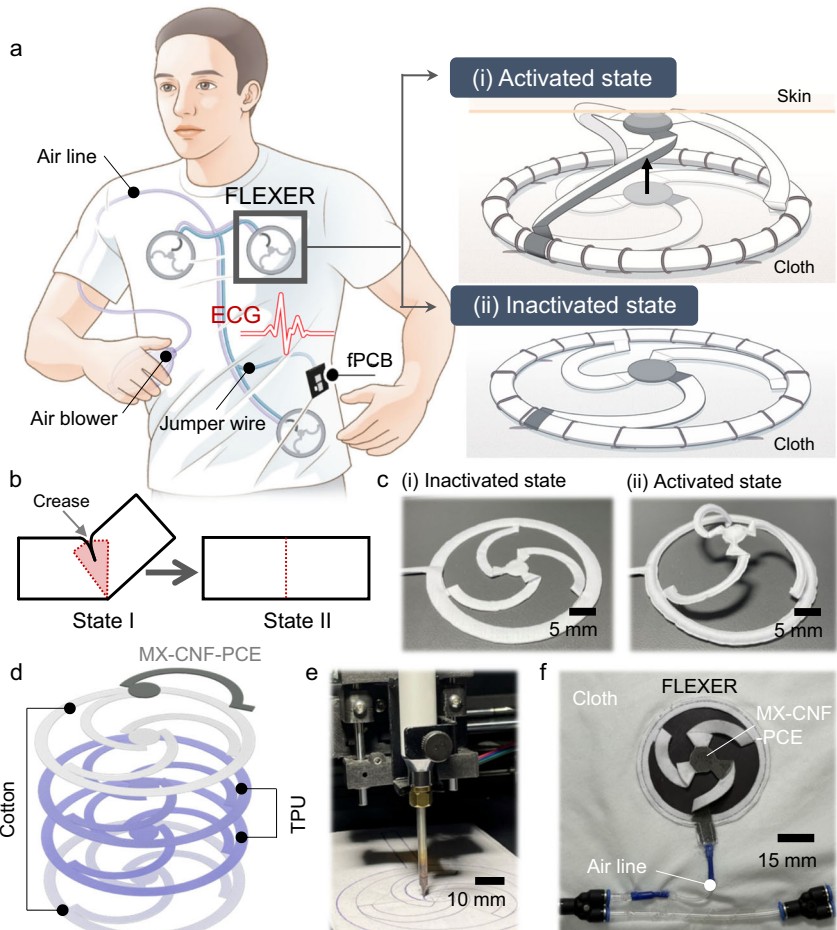

**Fig. 1 | Concept of FLEXER as a wearable device. a** Schematic illustration of ECG monitoring using controllable FLEXERs in (i) inactivated and (ii) activated states. **b** Activation process of Nesler's PIA joint in FLEXER. **c** Photograph of the inactivated and activated states of three-legged FLEXER. **d** Structural layout of FLEXER comprising MX-CNF-PCE, cotton, and TPU. **e** Photograph of the heat-sealing process using a customized XY plotter. **f** Photograph of backstitched FLEXER inside clothes.

providing a user-friendly and adaptable solution. The operation of the FLEXER is elucidated by Nesler's pneumatic interference actuator (PIA) model[24] (Fig. 1b). The PIA model can generate large deformation by strategically using the flexible structure's crease. Specifically, when the flexible air pocket is folded, it forms an overlapping volume at the creased section (Fig. 1b, State I). The overlapped volume decreases the overall volume of the air pocket below the original volume. Consequently, the air pocket tends to revert to its original volume by flattening the crease (i.e., unfolding) when subjected to pneumatic pressure (Fig. 1b, State II). During this process, the crease operates as a torque pivot point, inducing significant deformation in the structure[24,25]. The shape-morphing behavior of the FLEXER can be precisely controlled by coordinating multiple PIA joints. The symmetrical arrangement of double-folded PIA joints, with one end touching the anchoring frame and the other end touching the center, enables the controlled movement of the center electrode (CE) along an axis perpendicular to the contact plane of the FLEXER (Fig. 1c).

The FLEXER system comprises three key components, as depicted in Fig. 1d. The first component is the MXene/CNF/PCE (MX-CNF-PCE) composite electrode with a high electrical conductivity, flexibility, and oxidation resistance, and it is placed at the upper skin contact area. This electrode is seamlessly integrated into cotton fabric using a solution spray method. It leverages the intricate 3D woven structure of the fabric to form a conductive network of MXene in a 3D configuration. The 3D MXene electrode structure effectively confines mechanical damage to the outermost surface, thereby aiding in the preservation of conductivity in the event of electrode delamination due to adhesion failure[26]. (Supplementary Fig. 1) The second

component utilizes TPU, which is known for its mechanical resilience and stretchability, to create an air pocket that is a fundamental part of the PIA system within the FLEXER[27]. TPU films have a low melting point of 140–150 °C, making them easy to meld[28] using a customized XY plotter with a soldering iron. The TPU-laminated cotton fabric was precisely cut using a customized computer numerical control (CNC) cutting machine (Fig. 1e and Supplementary Figs. 2 and 3). Finally, the outer layer of the FLEXER is made from cotton fabric, which serves as its sheath. When users wear clothing embedded with the FLEXER, the overall appearance of the attire closely resembles regular clothing (Supplementary Fig. 4). This distinguishes it from other wearable devices that usually have a tighter fit[29]. Notably, the FLEXER remains inconspicuous externally, effectively reducing the possibility of social stigma associated with wearing it. The fabricated FLEXER was connected to an airline from a handheld air blower and stitched into the clothing (Fig. 1f). The advantages of the FLEXER also include structural simplicity, which ensures durability without intricate mechanical component requirements; rapid actuation due to friction-free pneumatic pressure; and paramount user safety, which stems from the absence of rigid parts and its PIA-based mechanism that allows for low pneumatic pressure usage and minimizes potential risks.

## Physical and design analysis of FLEXER

The straightforward linear structure with a double-fold design was examined to comprehensively explore the relationship between the pneumatic structure of the FLEXER and its performance and design parameters. The linear PIA, measuring 11.8 cm in length and 1.9 cm in width (see Fig. 2a, top photographs), underwent an unfolding

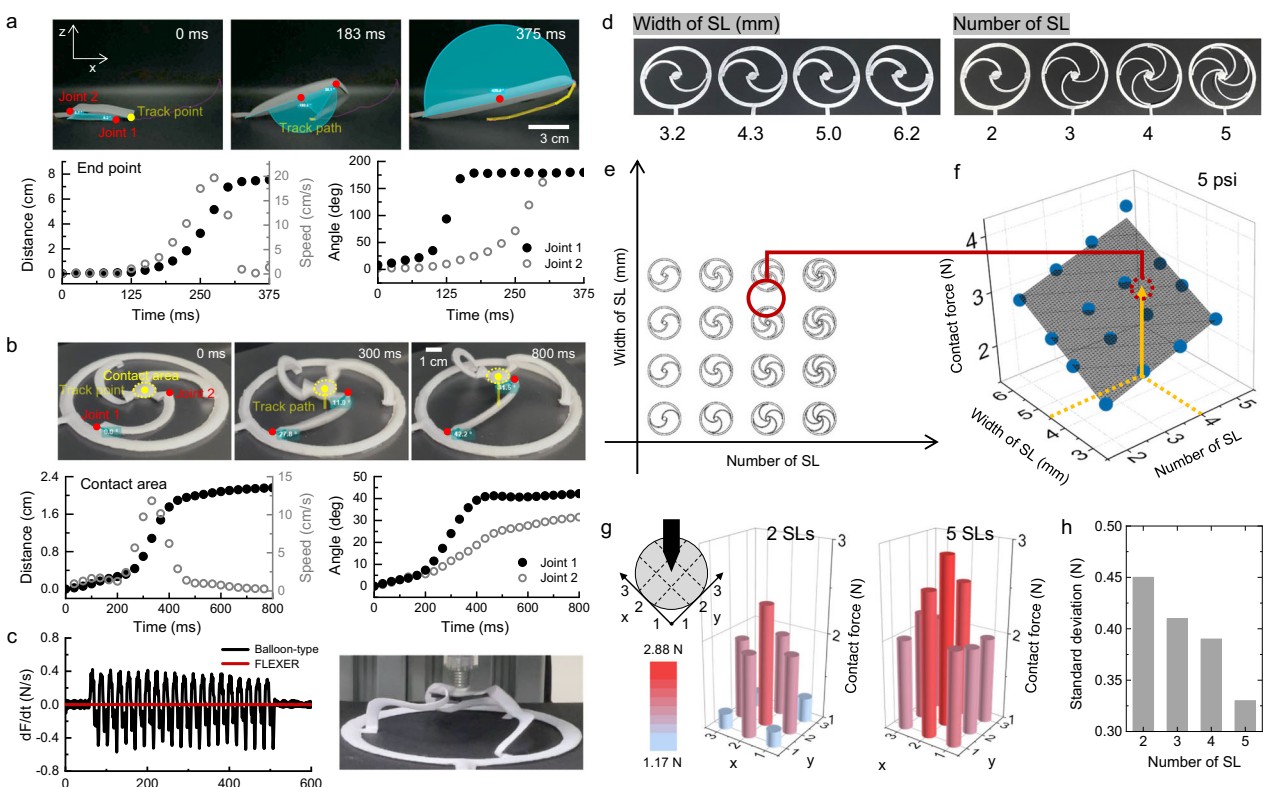

**Fig. 2 | Physical and design analysis of FLEXER.** Kinetic analysis of pneumatic actuation: Three actuation snapshots of **a** the straightened simple linear shape of FLEXER and **b** the three-legged FLEXER, along with corresponding distance, speed, and angle changes over time. **c** Contact stability measured by comparing the differentiation of force to time of the balloon-type and three-legged FLEXER. **d** Introduction of different SL widths and numbers in FLEXER. **e** Evaluation of 16 multi-legged symmetric FLEXER variants. **f** Multi-legged symmetric FLEXER

contact force based on an applied pneumatic pressure of 5 psi. The plane represents Eq. (1). **g** Force distribution of two- and five-legged FLEXERs on a 3 × 3 grid division of the contact area. **h** Standard deviation of the contact force for each partition based on the number of SLs of FLEXERs, referring contact force distribution data from Fig. 2g and Supplementary Fig. 13. Source data are provided as a Source Data file.

actuation, which was recorded and analyzed using video analysis software (Kinovea). The inherent material characteristics of the cotton fabric caused the linear PIA to exhibit a self-unfolding tendency even without the application of air pressure, as illustrated in Supplementary Fig. 5. To counteract this spontaneous unfolding, rubber magnets were affixed near the linear PIA joints using magnetic force to impede undesired expansion (see Supplementary Fig. 6). Thereafter, the movement of the linear PIA end point and the angle changes of its joints were observed during actuation. The graph in Fig. 2a (bottom right) illustrates that Joint 1 underwent rapid inflation and expansion between 100 and 125 ms, ultimately reaching a 180° angle by 175 ms (Supplementary Movie 1). Simultaneously, Joint 2 gradually expanded between 100 and 250 ms and then aggressively expanded at 250 ms after Joint 1 reached a straight angle. This behavior is probably due to the folded portions of the linear PIA weakly obstructing airflow, leading to a sequential inflation process. By 375 ms, all the joints had fully expanded to their maximum distance of 7.55 cm. Although the length of the linear PIA decreased from its initial value (11.8 cm) because of perpendicular expansion, the overall change in dimensions remained minimal (see Supplementary Fig. 7 and Supplementary Movie 2 for further clarification). The linear PIA actuation behavior revealed that the joint close to the air pressure application point tends to actuate faster than the joint at a further distance. This occurs because of the 180° bending angle of the airflow path that creates large drops of air velocity. Since the FLEXER has multiple semicircular legs that require synchronized actuation, the parts that transfer air pressure to each leg were designed in a circular shape to minimize airflow interference when air pressure is applied.

As discussed earlier, the folded self-intersecting sections within the air pouch joints function as springs, directing the unconstrained end of the air pouch to move in a manner that minimizes the volume of self-intersection. However, the actuation of a single-block air pouch actuator involves both lateral and vertical movements, making it difficult to precisely control the direction of desired inflation. This limitation necessitates the incorporation of multiple joints to effectively inhibit the lateral actuation of the air pouch. Our devised FLEXER addresses this necessity through a device architecture that encompasses (i) a CE that serves as an electric contact point, (ii) an anchoring frame (AF) that maintains the shape of the FLEXER and provides the necessary mechanical constraint for actuation, and (iii) multiple semicircular legs (SLs) strategically positioned in an equiangular arrangement to link the AF and the CE (Supplementary Figs. 8 and 9). As depicted in the top photographs in Fig. 2b, a three-legged FLEXER with a 10 cm AF diameter was fabricated to illustrate the design approach. The AF is securely affixed to the fabric, serving as an integral mechanical anchor for the entire structure. The dimensions of the CE can be flexibly adjusted by either being enlarged to enhance signal detection or reduced to minimize the overall size of the FLEXER. The SLs are linked to both the CE and AF through two pivotal joints. The first joint, which is situated at the AF, dictates the rotational direction of the SLs while functioning as a stabilizer at the CE level. Similarly, the second joint, which is located at the CE, serves as an electrode-level stabilizer, ensuring the stability of the entire system. When pneumatic pressure was applied, both joints unfolded, directing the CE vertically. While expansion, FLEXER's Joint 1 raised the SLs and CE by deflecting it from 0° to 42.2° (Fig. 2b, bottom right graph), and Joint 2 shifted the angle from 0° to 31.5°, ensuring parallel alignment between the CE and AF planes for stable skin contact. The FLEXER also achieved a rapid actuation speed of ~12 cm/s by deploying the CE in <0.4 s (Fig. 2b, bottom left graph). This rapid actuation proves advantageous in situations[30] where users require prompt analysis of vital signals (e.g., ECG and EMG signals) related to urgent medical conditions, such as coronary heart disease, cardiovascular disease, and congestive heart failure[31–34]. Furthermore, we assessed the contact stability of the FLEXER during its spring-like motion. The three-legged FLEXER was attached to a linear actuator, and a cyclic linear motion with a 20 mm travel length was applied to simulate dynamic movement while measuring the contact force of the CE. As a comparative measure, a balloon with the same diameter as the AF of the FLEXER was also tested (Fig. 2c and Supplementary Movies 3 and 4). The measured force was then differentiated with respect to time to quantify the change in force over time. The result revealed that the three-legged FLEXER exhibited a consistent time derivative of force during repetitive movement, highlighting its effectiveness in reducing motion-related interference during biosensing[35]. Contrastingly, the balloon displayed a standard deviation of the time derivative of force of 0.212 N/s, which is 80 times higher than that of the FLEXER (0.0026 N/s). This difference underscores the enhanced contact stability of the FLEXER in managing movement-related challenges during sensing operations. Moreover, we evaluated the deviation of the CE from its center after the activation of FLEXER because MXene was coated on only one SL of the FLEXER, extending from the SL to the AF as a conductive trace. Supplementary Figure 10 presents top-view images comparing the activation of MXene-coated and non-coated FLEXER samples. The CE diameter of the FLEXER is 10 mm, and the center of the CE remained aligned with the inner contact area after activation. This demonstrates the consistent movement of the FLEXER system along with the electrode materials.

To expand the FLEXER into a wider range of wearable bioelectrodes, the contact force of the electrodes must be adjustable. This can be achieved by selecting an appropriate shape and number of SLs. To evaluate the relationship between the FLEXER design parameters and the resulting contact force, a set of 16 distinct samples with varying SL widths and numbers were fabricated (Fig. 2d, e). Since the AF remained mechanically fixed and the movement of the CE depended on the behavior of the SLs, the design parameters for the FLEXER were confined to the shape and number of the SLs. The AF and CE diameters were maintained at 130 and 15 mm, respectively, while the pneumatic pressure was maintained at 5 psi to eliminate any additional control variable. Figure 2f presents the recorded contact forces of the different SL widths and numbers for the FLEXER. The data demonstrate that the contact force increased as the number and width of the SLs increased. This trend can be explained by the PIA mechanism, which establishes a connection between the output force and the volume of joint overlap. Regarding the FLEXER, the increase in the number and width of its SLs directly contributes to the enlargement of the overlapping areas. This increase in overlapping areas leads to a heightened contact force. Eq. (1) can be used to determine the contact force based on the two design parameters at a pneumatic pressure of 5 psi:

$$F = 0.25 + 0.35 * w + 0.24 * n \ (R^2 = 0.97), \tag{1}$$

where $F$ represents the contact force, and $w$ and $n$ represent the width and number of the SLs, respectively. This empirical relationship is graphically illustrated in Fig. 2f. The contact force is expressed as follows at high pneumatic pressures of 10 psi (Eq. (2)) and 15 psi (Eq. (3)) (Supplementary Fig. 11):

$$F = 0.26 + 0.33 * w + 0.30 * n \ (R^2 = 0.93), \tag{2}$$

$$F = 0.28 + 0.32 * w + 0.34 * n \ (R^2 = 0.90) \tag{3}$$

Additionally, the maximum contact distance of the FLEXER electrode depends on the FLEXER size, as shown in Supplementary Fig. 12. When the AF diameter of the FLEXER varied from 8 to 13 cm, the contact distance increased from 1.7 to 3.2 cm, all at a constant pressure of 10 psi. Considering both the magnitude and uniformity of the contact force is essential for the FLEXER to effectively function as a bioelectrode in accurately detecting biological signals. To assess the

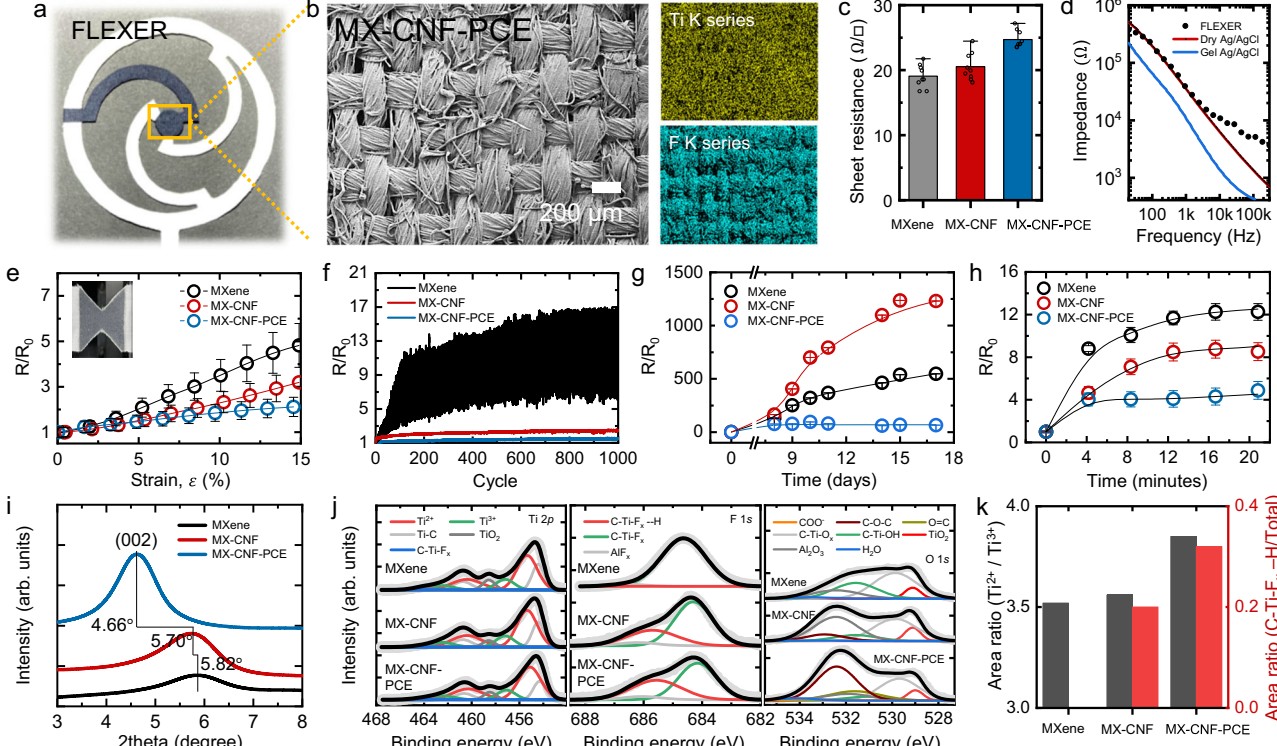

**Fig. 3 | Chemical analysis of the MX-CNF-PCE electrode. a** Photograph image of spray-coated MX-CNF-PCE of FLEXER. **b** SEM and EDX elemental mapping of MX-CNF-PCE. **c** Comparison of the sheet resistances of the MXene-based electrodes: pristine MXene, MX-CNF, and MX-CNF-PCE. Bar chart showing sample means ($n = 8$) with standard deviation error bars. **d** Skin impedance comparison between the reference dry and wet Ag/AgCl electrodes and pneumatically fully activated FLEXER. Normalized resistance stability measurement under four different conditions: **e** tensile strain showing sample means ($n = 5$) with standard deviation error bars, **f** 1000 cycles of 15% bending strain test, **g** 100% humidity environment showing sample means ($n = 5$) with standard deviation error bars, and **h** an artificial sweat droplet showing sample means ($n = 5$) with standard deviation error bars. **i** XRD measurement. **j** XPS graph. **k** XPS analysis results of $Ti^{2+}/Ti^{3+}$ area ratio and hydrogen bond ratio comparison referring from XPS deconvoluted peaks' area. Source data are provided as a Source Data file.

impact of contact force uniformity based on the number of SLs, the contact force of the CE was measured at various points using a $3 \times 3$ grid, as depicted in the schematic at the top-left corner of Fig. 2g. The contact force distributions of FLEXER models with two, three, four, and five SLs were analyzed, as shown in Fig. 2g and Supplementary Fig. 13. Calculating the standard deviation for FLEXER models with different numbers of SLs revealed the following trend: An increase in the number of SLs resulted in a more uniform distribution of contact force, as illustrated in Fig. 2h.

## Chemical analysis of MX-CNF-PCE electrodes

MXene was used as the electrode material of the FLEXER because of its exceptional electrical conductivity and compatibility with water-based solution methods. However, MXene has poor mechanical strength and experiences a reduction in electrical conductivity under ambient oxidation. To address these challenges, we incorporated PCE and CNF into the MXene solution to enhance the solution properties. When combined with MXene, PCE effectively envelops the MXene surface, providing a protective shield against oxygen exposure. Moreover, this combination enhances the flexibility of MXene by widening the gaps between its platelets. Conversely, CNF significantly increases the mesoscopic mechanical strength of the composite. Since MXene has a platelet-like structure, the interconnection of individual MXene platelets through covalent bonds can result in poor mechanical strength. CNF serve as a platform for MXene attachment because their dimensions (diameter: 3–30 nm and length: several microns) differ significantly from those of MXene (diameter: several microns and thickness: 1.2–1.8 nm[36]), facilitating the release of strain within the composite structure. Supplementary Figure 14 provides a schematic

representation of this composite, which encompasses both CNF and PCE elements. To realize this approach, a water-based solution containing MXene, CNF, and PCE (MX-CNF-PCE) was prepared. This MX-CNF-PCE solution was then spray-coated onto the contact side of the FLEXER structure using a polyethylene terephthalate (PET) shadow mask (Fig. 3a and Supplementary Fig. 15). The hydrophilic nature of the cotton fabric, the hydrophilic functional groups present on the MXene nanosheets, and the hydroxyl groups within the CNF and PCE all enabled the solution to be adsorbed onto the fabric, resulting in the formation of densely packed MX-CNF-PCE layers. We confirmed the uniformity and integrity of the MX-CNF-PCE spray-coating using scanning electron microscopy (SEM) and energy-dispersive X-ray (EDX) analysis[37], as illustrated in Fig. 3b. The top-view SEM image reveals that the MXene nanosheets enveloped the cotton fibers and bridged the gaps between them, while the unmodified cotton fabric exhibited a clean surface characterized by interwoven warp and weft yarns (Supplementary Fig. 16). Additionally, the uniform distribution of each element within the MXene-based solution was observed from the EDX mapping (Fig. 3b (right side) and Supplementary Fig. 17).

Generally, electrodes must have low sheet resistance to be able to measure biosignals with high reliability. Here, the MX-CNF-PCE and MX-CNF electrodes were measured to have sheet resistances of $24.7 \, \Omega/\square$ and $20.5 \, \Omega/\square$, respectively, which were slightly higher than that of the MXene electrode ($19.1 \, \Omega/\square$) (Fig. 3c). This is attributed to the inclusion of electrically nonconductive CNF and PCE within the MXene interlayers. However, their sheet resistances are still significantly lower than those of previously reported MXene-coated cotton electrodes ($569–761 \, \Omega/\square$)[37]. In addition to the sheet resistance of electrodes, the skin-to-electrode impedance is also a crucial factor. Skin-to-electrode

impedance reflects the skin's resistance in transmitting electrical signals from the user to the sensing element of the electrode. Low skin-to-electrode impedance is associated with high biosignal quality, as it signifies reduced resistance at the contact point between the skin and conductive surfaces[38]. Fig. 3d illustrates the measured skin-to-electrode contact impedances of the FLEXER electrodes and reference dry and gel-type Ag/AgCl electrodes. The inflated FLEXER electrode displayed skin-to-electrode impedance comparable to those of reference electrodes of the same size (diameter = 3 cm). Notably, the skin-to-electrode impedance of the FLEXER electrode was similar to that of the dry-type electrode, particularly up to a frequency of 2.45 kHz. At a frequency of 100 Hz, the FLEXER electrode exhibited an impedance of 191 kΩ, while the dry and gel Ag/AgCl electrodes exhibited impedances of 190 kΩ and 64 kΩ, respectively[39]. These impedance values are sufficiently low for accurate biosignal measurements without significant noise interference[40].

Further, we investigated the electromechanical responses of the MX-CNF-PCE bioelectrode under mechanical deformation. A 15% tensile strain was applied to the three electrodes (MXene, MX-CNF, and MX-CNF-PCE), and the normalized resistance ($R/R_0$) was measured, as depicted in Fig. 3e. MX-CNF-PCE showed the best electromechanical stability with the smallest $R/R_0$ increase of 2.1 at a 15% tensile strain, whereas MXene and MX-CNF exhibited $R/R_0$ values of 4.8 and 3.2, respectively. The enhanced stability of MX-CNF-PCE is attributed to its unique brick-and-mortar architecture. In this structure, the 2D MXene nanosheets function as bricks, while the 1D CNF serves as the mortar[41]. This is due to the formation of hydrogen bonds between the terminal groups (−OH, −O, and −F) of MXene and the hydroxyl groups of the CNF. The incorporation of PCE, a comb-type polymer, into MX-CNF-PCE introduces additional hydrogen bonds, which significantly enhance its mechanical stability. Owing to these improvements, MX-CNF-PCE exhibits a decrease in $R/R_0$ in stark contrast to MX-CNF. Furthermore, we conducted a cyclic bending stability test by subjecting the three MXene-based electrodes to a 15% strain repeatedly for 1000 cycles while measuring the $R/R_0$ values, as depicted in Fig. 3f. This test simulates situations where stress is frequently applied to the bioelectrode on the fabric overlying specific areas, such as joints like knees or elbows[42]. A consistent trend was observed throughout the test: The inclusion of CNF and PCE enhanced electromechanical stability. MX-CNF-PCE, MX-CNF, and MXene had $R/R_0$ values of 1.4−1.6, 2.3−2.6, and 6.1−16.9, respectively.

For the oxidation resistance test, the MXene-based electrodes were exposed to a sealed glass chamber with 100% relative humidity for 17 days, as illustrated in Supplementary Fig. 18. The results of this rigorous environmental test revealed the remarkable oxidation stability of MX-CNF-PCE, as evidenced by a modest increase in its $R/R_0$ value from 1 to 65 (Fig. 3g). This marginal increase in $R/R_0$ became apparent when compared to those of the MXene and MX-CNF electrodes[43], which experienced substantial increases to 1231 and 548 (20 times and 9 times that of MX-CNF-PCE), respectively. As mentioned earlier, this is attributed to the protective barrier formed by the incorporation of PCE, which effectively shields MXene from oxidation. Sweat excreted by the skin can also affect the performance of bioelectrodes, thereby affecting their $R/R_0$ ratio. To assess the stability of MX-CNF-PCE in response to perspiration, its electrical characteristics were observed in an artificial sweat[44]. After wetting the three MXene-based electrodes with the artificial sweat (see Supplementary Fig. 19), we monitored their electrical response ($R/R_0$) for 20 minutes (Fig. 3h). MX-CNF-PCE exhibited the most electrically stable response because of the PCE coating on the MXene nanosheets. Additionally, experiments were conducted to determine the optimal mixing ratio of MX-CNF-PCE for enhanced electromechanical properties by doubling and halving the CNF and PCE amounts, as shown in Supplementary Fig. 20. Since CNF and PCE act as insulators for the electrode, their proportion needs to be minimized in comparison to MXene. Considering the spray-coating

capability and the electrical conductivity of the bioelectrode, we determined the optimized mass ratio of MXene to be 82 wt% of the total electrode composition. Maintaining this MXene weight ratio (82 wt%), we precisely adjusted the amounts of CNF and PCE in the electrode to further improve its environmental stability. By comparing the environmental stability of bioelectrodes with different CNF and PCE ratios, we identified the weight ratio of 24:4:1 (MXene:CNF:PCE) as optimal for the FLEXER system. Consequently, the MX-CNF-PCE electrode demonstrated enhanced electromechanical and environmental resistance properties under various mechanical deformations, including tensile strain and cyclic bending, 100% humidity environment, and interference from sweat.

Spectroscopic characterization was performed on the MX-CNF-PCE electrode to understand the chemical structure underlying its improved electromechanical test results (Fig. 3i–k). The X-ray diffraction (XRD) curves in Fig. 3i and Supplementary Fig. 21 reveal evidence of the intercalation and blending of the CNF and PCE within the MXene nanosheets. Specifically, the (002) diffraction peak for MXene appeared at 2θ = 5.82°, which shifted to 5.70° in MX-CNF and 4.66° in MX-CNF-PCE. These shifts correspond to an increase in the interlayer spacing from 3.04 nm to 3.10 nm and 3.78 nm, respectively[22,45–47]. Fig. 3j presents the X-ray photoelectron spectroscopy (XPS) spectra of the pristine MXene, MX-CNF, and MX-CNF-PCE composite films. An analysis of the Ti $2p$ and O $1s$ spectra reveals that the area ratio of $Ti^{2+}$ to $Ti^{3+}$ peaks increased from 3.52 (MXene) to 3.56 and 3.85 (MX-CNF and MX-CNF-PCE, respectively). Simultaneously, the area ratio of the −COO− peak to the total chemical bonds decreased from 0.04 (MXene) to 0.02 (MX-CNF and MX-CNF-PCE). These observations confirm the formation of covalent Ti−O bonds between the MXene nanosheets and the PAA anchor of PCE, as revealed by the Ti $2p$ spectrum (i.e., −COOH). Furthermore, the F $1s$ spectra reveal the presence of hydrogen bonds between the MXene nanosheets and the hydroxyl groups in the CNF or the partially hydrolyzed −COO− of PCE (i.e., −COOH). The area ratio of the hydrogen bond peak (C−Ti−$F_x$−H) to the total area increased from 0 (MXene) to 0.2 and 0.32 (MX-CNF and MX-CNF-PCE, respectively). The appearance of a high binding energy peak in the F $1s$ region for C−Ti−$F_x$−H demonstrates the formation of hydrogen bonds, which contributed to MX-CNF and MX-CNF-PCE exhibiting better mechanical stability than pristine MXene. These ratio increments are depicted in Fig. 3k.

## Wireless electrophysiological monitoring using FLEXER

To showcase the capabilities of the FLEXER in wearable biosensing, we developed a wireless electrophysiological monitoring system. Figure 4a provides an overview of the working concept and block diagram of the developed system (see Supplementary Figs. 22 and 23 for more details). The FLEXER and the flexible printed circuit board (fPCB) were electrically connected through jumper wires, allowing electrophysiological signals detected by the FLEXER to be routed to the biopotential amplifier on the fPCB. These signals were then amplified and filtered by the biopotential amplifier before undergoing analog-to-digital conversion (ADC). The converted signals were transmitted wirelessly to the computing unit in real time via a Bluetooth Low-Energy (BLE) module for further analysis. An ECG measurement was conducted with the developed wireless electrophysiological monitoring system (Fig. 4b, see Supplementary Fig. 24 left for details). As a control group, ECG signals were also measured using commercially available gel-type and dry Ag/AgCl electrodes with the developed wireless system. To record the ECG signals, two electrodes were placed on the left and right sides of the chest, and a ground electrode was placed on the lower abdomen[48]. During the ECG recording, the FLEXER was affixed to the clothing, while both gel-type and dry Ag/AgCl electrodes were securely attached to the skin. Despite the gap between the clothing and the skin, the FLEXER exhibited a high signal-to-noise ratio (SNR) because of its unique

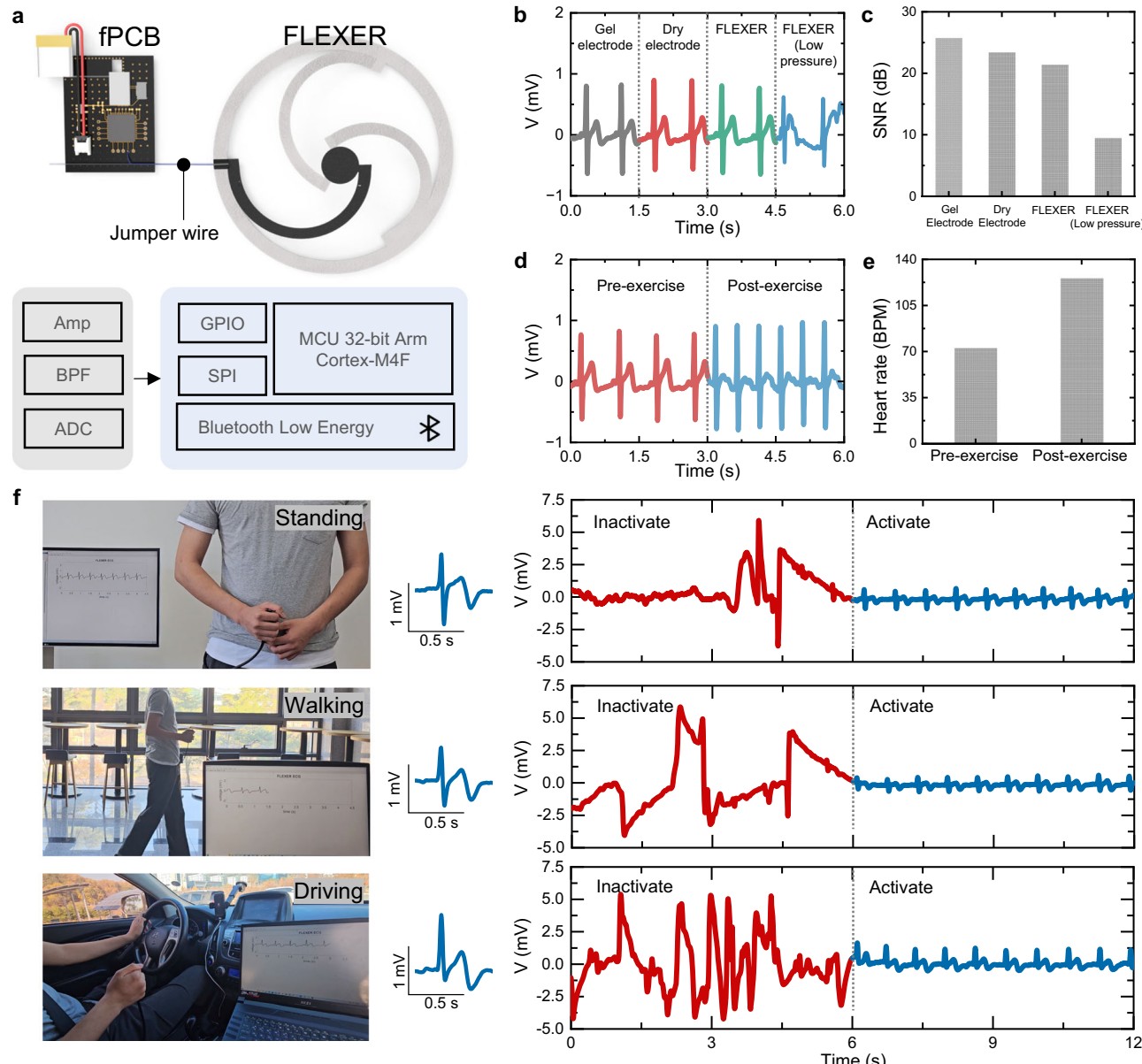

**Fig. 4 | Real-time ECG monitoring using FLEXER. a** Schematic illustration and block diagram of the wireless electrophysiological monitoring system using FLEXER. **b**, **c** Comparisons of the representative ECG signals (V, voltage) (**b**) and their SNR values recorded by various electrodes (**c**). **d**, **e** Representative ECG signals (V, voltage) (**d**) and heart rate (**e**) obtained before and after exercise. **f** Real-time measurement of ECG using multi-FLEXER in situations involving standing, walking, and driving, both in the inactive and active status of FLEXER. Source data are provided as a Source Data file.

pneumatic structure, which facilitated effective skin-electrode contact (Fig. 4c). Conversely, the FLEXER with low air pressure displayed a low SNR owing to inadequate skin-electrode contact. Consequently, the PQRST waveforms in the ECG recorded with the FLEXER closely resembled those obtained using commercial Ag/AgCl gel electrodes (Supplementary Fig. 24 right). Moreover, the FLEXER demonstrated its efficacy in monitoring changes in heart rate via ECG measurements performed before (72 BPM) and after (125 BPM) exercise, as shown in Fig. 4d, e. Unlike commercial Ag/AgCl gel electrodes, which necessitate continuous skin contact for long-term monitoring, FLEXERs only make contact with the skin during measurements. This minimizes discomfort even during extended periods of use, which is crucial for ensuring user comfort and compliance in continuous health monitoring scenarios. Supplementary Movie 5 shows emergent status of single-FLEXER and multi-FLEXER when pneumatic force is occurred which can contact to skin with facile

hand pumping. Real-time measurement of ECG using multi-FLEXER in situations involving standing, walking, and driving, both in the inactive and active status has been also proceeded in Fig. 4f. During real-time ECG monitoring, signals were dramatically got stabilized after the activation of FLEXER. (Supplementary Movie 6)

The FLEXER is also highly suitable for capturing EMG signals, which are foundational for hand gesture recognition, because of its enhanced electrical conductivity and stable skin contact due to its pneumatic-based design. This capability has profound implications in areas such as human-machine interfaces and particularly prosthesis control in healthcare[45,46,49]. Gesture recognition via the EMG process entails the placement of multichannel electrodes on various areas of the arm, with recognition being facilitated by extracting EMG signal features using various classifiers (Fig. 5a)[47]. To demonstrate this, EMG signals corresponding to eight distinct gestures were measured using FLEXERs positioned on the forearm, as depicted in Fig. 5b. The

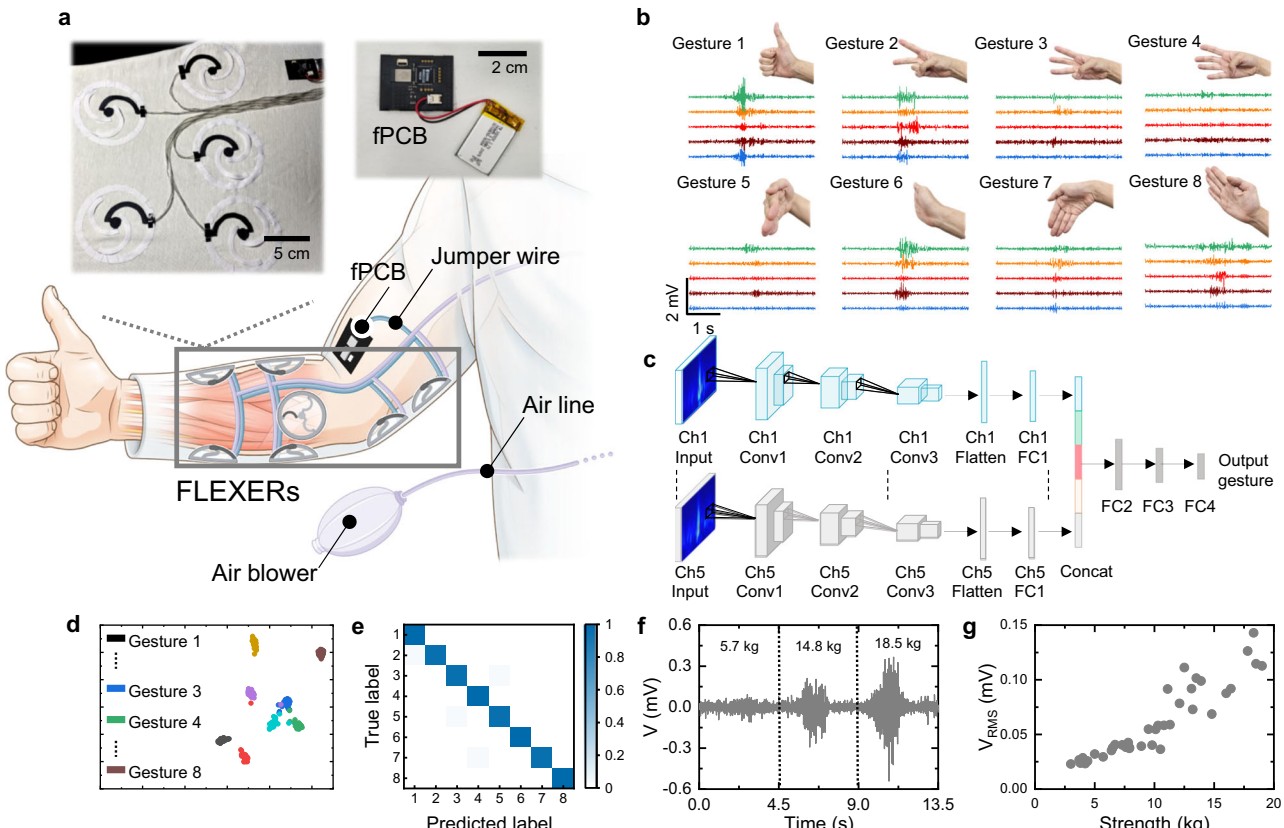

**Fig. 5 | EMG monitoring using FLEXER. a** Conceptual illustration of EMG monitoring using FLEXER. **b** Hand gestures for gesture recognition and corresponding EMG signals. **c** Pipeline of multi-input CNN architecture for gesture recognition. **d** t-SNE visualization of the gesture recognition (Fold 5). **e** Confusion matrix of the gesture recognition result (Fold 5). Each number on the axis of the confusion matrix corresponds to a unique gesture. **f** EMG signals during the grip strength test. **g** $V_{RMS}$ with different handgrip strengths. Source data are provided as a Source Data file.

recorded EMG signals, captured across five channels for each gesture, displayed distinct patterns among the gestures. The signals were then subjected to the continuous wavelet transform (CWT), which transformed them into scalograms, i.e., image-like time-frequency representations. These scalograms served as inputs for a multi-input convolutional neural network (CNN)-based gesture recognition system[50]. For each gesture, the scalograms derived from the five channels underwent independent processing through three convolution layers and one fully connected layer to extract unique features. These processed outputs were then concatenated to create a comprehensive feature set for each gesture. This concatenated layer was subsequently linked to three additional fully connected layers to classify the eight gestures (Fig. 5c). Further details can be found in Supplementary Table 1 and Supplementary Figs. 25 and 26. To evaluate the generalizability of the system across independent datasets, five-fold cross-validation tests were performed on randomly mixed datasets. The high-dimensional features extracted from the multi-input CNN model were visualized using t-distributed stochastic neighbor embedding (t-SNE). This visualization displayed a distinct separation of features, contributing significantly to the achievement of an accuracy rate of 97.92% in Fold 5 (Fig. 5d and Supplementary Fig. 27). The recognition accuracy using CNN model for all eight gestures in each fold is presented as a confusion matrix (Fig. 5e and Supplementary Fig. 28). The average accuracy across the five-fold cross-validation tests was 95.94%. This high accuracy is mainly due to the aforementioned SNR of the FLEXER. Generally, EMG signals can be used to assess grip strength[15,51]. Thus, a grip strength test was conducted by performing gripping actions at various force levels (Fig. 5f). During this procedure, a dynamometer quantified the force level, and the corresponding EMG signals were recorded using the FLEXER on the flexor carpi radialis of the forearm (Supplementary Fig. 29). The recorded EMG signals were then segmented into two-second time windows, and the root mean square (RMS) of the signals were calculated. As depicted in Fig. 5g, the RMS of the EMG signals exhibited a corresponding increase with grip strength.

## Discussion

In conclusion, the development of FLEXER, a lightweight and flexible wearable bioelectrode with a pneumatic structure, represents a significant advancement in real-time healthcare signaling technology for wearable devices. The incorporation of MXene-based dry electrodes allows the FLEXER to achieve exceptional electrical conductivity and significantly reduce skin-to-electrode contact impedance, thereby addressing key challenges in biosensing applications. Moreover, the innovative design of the FLEXER ensures consistent skin contact, enabling stable signal recording and accurate monitoring of physiological parameters. Comprehensive evaluations demonstrated the impressive electromechanical properties of the FLEXER, which include exceptional mechanical stability, strain resistance of up to 15%, and remarkable durability over 1000 bending cycles. Additionally, the real-time wireless electrophysiological monitoring system developed using the FLEXER enables seamless recording and transmission of ECG and EMG signals. Furthermore, the FLEXER demonstrates exceptional accuracy in monitoring curved regions, with a recognition accuracy of 95.94% for hand gesture recognition using EMG signals. These significant advancements highlight the potential of FLEXERs for personalized healthcare applications, such as prosthesis control and rehabilitation, and underscore their transformative impact on the field of wearable bioelectrodes.

## Methods

### Fabrication of FLEXER

The FLEXER design was simplified using the visual programming software Rhino 7 and Grasshopper 3D (Supplementary Fig. 9). Two pieces of TPU-laminated cotton fabric were overlapped, ensuring that the TPU layers came into direct contact. Hot sealing with a pen-type soldering tip was conducted along the designed pathway using a customized commercial XY plotter (Liyang equipment), and the fabric was cut to the desired size using Cameo 4 pro (Silhouette). After preparing the pneumatic structure of the FLEXER, the MX-CNF-PCE solution was spray-coated on it through a PET shadow mask to create a conductive path on the FLEXER. Pneumatic force was applied to the FLEXER through an air blower connected to an airline.

### Synthesis of $Ti_3C_2T_x$ MXene and MXene-based electrodes

The etchant solution was prepared in a 250 mL Teflon beaker as follows: 3.2 g of LiF (99.9%; Sigma-Aldrich) was dissolved in 40 mL of a 9 M HCl aqueous solution and cooled to <5 °C using an ice-water bath. The etchant was then deoxygenated with ultrahigh-purity $N_2$. Subsequently, 2.0 g of MAX phase ($Ti_3AlC_2$ powder, 99%, 325 mesh; Forsman) was gradually added to the etchant under gentle stirring. The mixture was deoxygenated again and protected in an inert atmosphere. Etching was performed in an oil bath at 25 °C for 48 h to remove most of the Al layer from the MAX phase. The resulting product was washed and delaminated in two 50 mL centrifuge tubes using deoxygenated water and then subjected to five to six cycles of centrifuge-shaking (Labogene 416, ONE Commercial shaker Horizontal Single Head Shaking Machine 220 V LJS80-1). Finally, the delaminated MXene was sonicated in an ice bath for 30 min and centrifuged at 2826×$g$ for 30 min to obtain a supernatant predominantly composed of single-layer MXene. The MXene dispersion was mixed with a CNF polymer aqueous solution (1 wt%; CNNT) and a PCE polymer aqueous solution ($M_W$ = 50 k, 55.0 wt %; L'BESTE GAT) according to the mass ratio specified in the experimental design (MXene:CNF:PCE = 24:4:1, 24:2:1, 24:4:2, and 24:2:2). The mixture of MXene and polymers was vigorously shaken for 30 min to obtain a homogeneous MXene–polymer composite material.

### Characterization

All video analysis of the pneumatic structure of the FLEXER was performed using the Kinovea program. The contact force of the FLEXER was measured using a digital force gauge (ZTA-5N; IMADA) under various pressure and FLEXER design conditions. The surface morphologies and elemental distribution of the MXene-based electrodes were characterized using SEM (JSM-7610F-Plus) and EDX analyses. The sheet resistance was measured using the four-point probe method (Keithley 2182A, 2634B, and 6221). The skin-to-electrode impedance was measured using an inductance–capacitance–resistance (LCR) meter (E-4980AL; KEYSIGHT) in the frequency range of 20 Hz to 300 kHz. Contact stability and tensile strain stability tests were conducted to evaluate the mechanical robustness of the MXene-based electrodes on the FLEXER using a stepper motor controller (ECOPIA, BM-111). All electromechanical resistance data of the MXene-based electrode were gathered using a parameter analyzer (4200A-SCS; Keithley). XRD (SmartLab; Rigaku) was used to detect the differences in the interlayer spacing of the MXene-based electrodes. An X-ray photoelectron spectrometer (K-alpha, Thermo U.K.) was operated using an aluminum anode as the source with a spot size of 400 μm.

### Wireless electrophysiological monitoring

An electrophysiology amplifier (RHD 2132, Intan Technologies) with a 16-bit ADC and a differential gain of 192 was used to amplify, filter, and digitize electrophysiological signals. The sampling frequency was 258 Hz, and the bandpass filter frequency was 0.5–100 Hz. A BLE system on chip (QN9080SIP; NXP Semiconductors) wirelessly transmitted the data to a recording laptop for further analysis. The entire system on the fPCB was powered by a 3.7 V lithium polymer battery and electrically connected to the FLEXER via jumper wires (AWG 30).

### Real-time heart rate measurement using ECG

The ECG was measured by two electrodes placed on the chest and a ground electrode placed on the lower abdomen. In addition to the FLEXER, gel-type (Vitrode L-150X; Nihon Kohden) and dry (Laxtha) Ag/AgCl electrodes were used for noise comparison of the signals. The Ag/AgCl dry electrodes were attached to the skin using medical tape (2702; 3 M). The commercial electrodes were connected to the developed fPCB via snap connectors (Laxtha). The recorded ECG signals were transmitted wirelessly to the laptop through the BLE system on chip of the developed wireless monitoring system. A fourth-order Butterworth band-pass filter (0.5–35 Hz) was used to denoise artifacts such as motion or power line artifacts. Subsequently, the R peak was detected to calculate the heart rate in beats per minute. The SNR of a single PQRST wave in the ECG signals was calculated as follows:

$$SNR_{dB} = 20 \log_{10} \frac{A_{PQRST}}{A_{noise}}, \qquad (4)$$

where $A_{PQRST}$ and $A_{noise}$ are the amplitudes of the PQRST and noise, respectively.

Thereafter, the SNR was averaged over the PQRST waves in the recorded ECG signals. The signal processing was performed using MATLAB 2022b (MathWorks). The real-time ECG was measured with the subject standing, walking, and driving, in both the inactive and active statuses of multi-FLEXER.

### Gesture recognition and grip strength test using EMG

For gesture recognition, five FLEXERs were placed on the forearm muscle. In addition, a FLEXER was placed on the flexor carpi radialis for grip strength test. Both the ground and reference electrodes were positioned proximally to the elbow. Thereafter, the recorded EMG signals were transmitted wirelessly to the laptop using the aforementioned system. Power line interference in the EMG signals was removed via a 60 Hz notch filter, and the signals were subsequently filtered using the fourth-order Butterworth band-pass filter (10–80 Hz). Gesture recognition was conducted on 120 EMG signal datasets for each eight distinct gestures. In these datasets, the signals were z-normalized and then segmented using a 4.5 s time window. Subsequently, scalograms were extracted through the CWT using the Morlet wavelet to provide image representations that could serve as input for a multi-input CNN-based classifier. The scalogram image was then resized to 64 × 64 and normalized. The input size was set at 64 × 64 × 3 to account for the RGB layers within each image. The architecture of the model comprised three convolution layers and one fully connected layer per channel, which were subsequently concatenated by a concatenation layer. This concatenated layer was connected to three additional fully connected layers and was ultimately followed by a Softmax layer for the recognition of the eight gestures. A rectified linear unit (ReLU) was selected as the activation function for the model. Meanwhile, the Adam optimizer (learning rate = 0.0001) was used as an optimizer, and loss consumption was performed using the cross-entropy loss function. Five-fold cross-validation was performed for generalized performance. After training for the multi-input CNN for 100 epochs, the accuracy of the five-fold cross-validation was averaged. The grip strength test was conducted using a dynamometer, with the volunteer maintaining a fixed elbow position. The dynamometer measured the maximum force level as the volunteer applied various grip strengths with the right hand. EMG signals were simultaneously recorded by the FLEXER on the flexor carpi radialis. These EMG signals were segmented using a 2 s time window to ensure that all grip-related signal variations were included. The RMS value of the segmented EMG signals was computed to establish a correlation between the maximum force level and the

RMS value derived from the grip-induced EMG signals. All signal processing was performed using MATLAB 2022b (MathWorks), and deep learning for gesture recognition was performed using TensorFlow (Google).

### Ethics information for human subject research

The authors confirmed that under Article 13, Paragraph (1) of the Enforcement Rule of Bioethics and Safety Act of the Ministry of Health and Welfare, Korea, Institutional Review Board approval was not needed as volunteers used wearable sensors with simple contact measurement. In total of 2 participants were male and average age was 27 years old. The authors affirm that human research participants provided informed consents for publication of the images in Fig. 4f and Supplementary Figs. 4, 25, and 29.

### Statistics and reproducibility

The SEM images in Fig. 3b and Supplementary Figs. 16 and 17 were tested with each three samples and five times repeated for each sample to get similar results.

### Reporting summary

Further information on research design is available in the Nature Portfolio Reporting Summary linked to this article.

## Data availability

All data supporting the findings of this study are available within the article and its supplementary files. Any additional requests for information can be directed to, and will be fulfilled by, the corresponding authors. Source data are provided with this paper.

## Code availability

The custom codes used in this study for wireless electrophysiology monitoring and signal analysis are available from the corresponding author upon request.

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

## Acknowledgements
This work was supported by the National Research Foundation of Korea (NRF) grant funded by the Korean government (MSIT) (No. RS-2023-00234581 and No. RS-2024-00415347) and the Korea Medical Device Development Fund grant funded by the Korean government (the Ministry of Science and ICT, the Ministry of Trade, Industry and Energy, the Ministry of Health & Welfare, and the Ministry of Food and Drug Safety) (Project Number: 1711174497, RS-2020-KD000093).

## Author contributions
Conceptualization: S.L. and D.H.H. Experiment: S.L., D.H.H., and J.J. Methodology: S.L., D.H.H., and J.J. Main data analysis and manuscript writing: S.L., D.H.H., and J.J. Assisted data analysis and manuscript writing: S.Y.C., Y.J.C., S.O., and Y.Y.C. Visualization: S.L., D.H.H., and J.J. Supervision: T.L., K.-I.J., and J.H.C. All authors contributed to the writing of the manuscript. All authors have approved the final version of the manuscript.

## Competing interests
The authors declare no competing interests.
