## [Peer Review File · Nature Communications]

REVIEWER COMMENTS

Reviewer #1 (Remarks to the Author):

Authors reported fabric-based lamina emergent MXene-based electrode for electrophysiological monitoring. The characterization and analysis are sufficient to this topic. The covered research in this article is worth to publish. However, in its current state, the manuscript does not meet the desired standard. I have several comments to the authors.

1. The introduction explains the problems with a range of sensors and emphasizes the need for wearable devices to be comfortable to wear, yet the device in this work does not seem to be very portable or comfortable to see.
2. Line 96, how it proved to be the complex weave structure of the fabric that enhanced the adhesion between MX-CNF-PCE and the substrate? The 100-cycle test in Figure 3f is also not enough to show its stability and more cycles need to be added.
3. Why the weight ratio of MXene-CNF-PCE was chosen to be 24:4:1 from Supplementary Fig. 17?
4. According to the Bragg diffraction formula, there is a problem with the calculation of layer spacing in XRD, and the authors should give a wide range of XRD patterns. In addition, the authors need to prove their structure with detailed tests and add the reason for their increased interlayer spacing?

Reviewer #2 (Remarks to the Author):

The authors have developed a fabric-based MXene electrode (FLEXER), which aims to improve comfort in traditional wearables. FLEXER offers excellent conductivity, low impedance, and shape-morphing capability, making it particularly suitable for dynamic users. The inclusion of pneumatic actuation as a novel mounting method for ECG and EMG biosensing is highly interesting. Therefore, I recommend this paper for publication in Nature Communications after necessary revisions.

1. The MX-CNF-PCE solution was applied to one leg of the FLEXER structure, leaving the other two legs uncoated. This asymmetrical coating may cause the center electrode to deviate from its intended perpendicular axis relative to the FLEXER's contact plane. Testing and an explanation of this issue are recommended.

2. On line 155, Page 5, the authors claimed that "The FLEXER also achieved a rapid actuation speed of ~20 cm/s by deploying the CE in < 0.2 s (Fig. 2b)". However, upon examining the data (both distance and angle) presented in Figure 2, it becomes evident that it would take at least 400 ms for the FLEXER to fully deploy the CE.

3. The author should conduct tests on the electrical response of their bioelectrodes using artificial sweat instead of a saline solution. This is because the ingredients of sweat and saline solution are significantly different, and sweat can potentially impact the performance of the electrodes.

4. The authors showcase FLEXER's effectiveness in wearable biosensing during motion, but concerns remain about its performance during large movements and compatibility with looser clothing. Further explanation is recommended.

5. Only 16 samples are displayed in Figure 2e. It is suggested to remove the three sets of ellipses.

6. The top row pictures in Figure 2a lack clarity. It is advised to avoid using a black background.

Reply to Reviewer #1

Authors reported fabric-based lamina emergent MXene-based electrode for electrophysiological monitoring. The characterization and analysis are sufficient to this topic. The covered research in this article is worth to publish. However, in its current state, the manuscript does not meet the desired standard. I have several comments to the authors.

→ We appreciate the comments from the reviewer. We have diligently revised the manuscript in response to these comments to further elaborate and consolidate the concept of FLEXER. Additionally, we conducted further experiments to clarify any uncertain points and have updated the manuscript accordingly.

1. The introduction explains the problems with a range of sensors and emphasizes the need for wearable devices to be comfortable to wear, yet the device in this work does not seem to be very portable or comfortable to see.

→ Thank you for the comment. Current wearable healthcare technologies that measure bioelectrical signals, such as ECG, EMG, or EEG, require direct contact with the skin to measure changes in impedance. For this reason, the conventional systems mentioned in the introduction need consistent external pressure to maintain skin contact. This can cause an abnormal sensation, leading to user discomfort, which is a significant issue for daily wearable devices. Moreover, as the focus of wearable healthcare systems shifts towards the ubiquitous measurement of biosignals, seamless integration into an individual's daily life is more important than ever. FLEXER is a system that can be integrated into the normal clothing we wear every day, eliminating the need for the tight clothing that many wearable healthcare devices require for consistent pressure. The shape-morphing characteristics of FLEXER avoid the need for consistent external pressure, making it a comfortable, non-intrusive solution for the wearable healthcare system.

Furthermore, FLEXER represents a disruptive technology for a transformable bioelectrode, currently positioned at the 3rd level of Technology Readiness Level (TRL). Through subsequent phases of development and design, improvements in FLEXER packaging and the use of diverse materials are anticipated to improve wearability, forming our blueprint to advance FLEXER in subsequent research and development endeavors.

2. Line 96, how it proved to be the complex weave structure of the fabric that enhanced the adhesion between MX-CNF-PCE and the substrate? The 100-cycle test in Figure 3f is also not enough to show its stability and more cycles need to be added.

→ We value the reviewer's insightful feedback. Our primary intention was to illustrate how the intricate 3D woven pattern reduces physical strain and friction. However, it seems our description may have come across as overstated. The combination of the cotton weaving structure and the spray-coated MX-CNF-PCE electrode material enhances physical durability compared to that on a flat substrate. Additionally, the hygroscopic nature of the cotton fabric facilitates solution absorption into each small strand, aiding in the formation of a robust and stable electrode structure. To demonstrate this enhanced durability, we conducted tape tests involving the attachment and detachment of adhesive tape on MX-CNF-PCE electrode layers deposited on both cotton and flat PET substrates to evaluate the adhesion stability of the electrode. (See **Supplementary Fig. 1.**) Following the tape test, the normalized resistance of the MX-CNF-PCE spray-coated layer on cotton showed an increase by 1.87 times, whereas on the PET substrate, it surged by 24.5 times, illustrating a significant contrast in adhesion behavior. (Refer to PNAS, **106**, 21490-21494, 2022.) This indicates that the 3D conductive network can effectively limit chemical and physical damage to the outermost layer of the fabric electrode, thereby preventing the loss of electrical conductivity of the conductive fillers within the fabric structure. Furthermore, we conducted a 1000-cycle test to further assess the mechanical stability, and the stability trend remained consistent. We have updated the supplementary information and the main manuscript to include this information.

Supplementary Fig. 1. Tape test comparing PET substrate and cotton fabric with a complex weave structure. (a) Normalized resistance changes in the conductive traces of PET and cotton fabric before and after the tape test. **(b)** Photographs of the spray-coated PET and cotton fabric before and after the tape test.

Line 97-101: This electrode is seamlessly integrated onto cotton fabric using a solution spray method. It leverages the intricate 3D woven structure of the fabric to form a conductive network of MXene in a 3D configuration. The 3D MXene electrode structure effectively confines mechanical damage to the outermost surface, thereby aiding in the preservation of conductivity in the event of electrode delamination due to adhesion failure.²⁶ **(Supplementary Fig. 1)**

Line 262-264: Furthermore, we conducted a cyclic bending stability test by subjecting the three MXene-based electrodes to a 15% strain repeatedly for 1000 cycles while measuring the R/R_0 values, as depicted in **Figure 3f**.

Line 265-267: A consistent trend was observed throughout the test: The inclusion of CNF and PCE enhanced electromechanical stability. MX-CNF-PCE, MX-CNF, and MXene had R/R_0 values of 1.4-1.6, 2.3-2.6, and 6.1-16.9, respectively.

Fig. 3 | Chemical analysis of the MX-CNF-PCE electrode. Normalized resistance stability measurement under four different conditions: (f) 1000 cycles of 15% bending strain test

3. Why the weight ratio of MXene-CNF-PCE was chosen to be 24:4:1 from Supplementary Fig. 17?

→ The preparation process for all electrode materials was evaluated based on their electrical conductivity, physical stability, and the feasibility of spray-coating. Each component plays a crucial role in the performance of bioelectrodes. MXene was chosen as the electrode material for FLEXER due to its outstanding electrical conductivity and compatibility with water-based solution methods. However, MXene exhibits poor mechanical strength and a decrease in electrical conductivity upon exposure to ambient humidity. To solve these issues, we introduced PCE and CNF into the MXene solution to improve its environmental resistivity. PCE effectively covers the MXene surface, providing protection against oxygen exposure while also

enhancing its flexibility by increasing the spacing between its platelets. On the other hand, CNFs significantly enhance the mesoscopic mechanical strength of the composite.

Since CNF and PCE act as insulators for the electrode, it was necessary to minimize their proportion. We determined an optimized mass ratio for MXene, which is greater than 82 wt% of the total electrode composition, by considering the fabrication system and the electrical conductivity of the bioelectrode. To achieve this, we varied the weight ratio of CNF and PCE by either halving or doubling it, resulting in MXene:CNF:PCE ratios of 24:4:1, 24:2:1, 24:4:2, and 24:2:2. Upon comparing the physical stability of these final four bioelectrodes, we concluded that the 24:4:1 MX-CNF-PCE ratio was the most stable composition, as demonstrated in **Supplementary Fig. 17**. We revised the legends in **Supplementary Fig. 17** to detail the ratios of each electrode material for clearer comprehension. Additionally, diluting CNF and PCE led to a solution with lower viscosity, which facilitated the formation of smaller droplets during spraying and ensured a uniform coating on the substrate.

Line 280-288: Additionally, experiments were conducted to determine the optimal mixing ratio of MX-CNF-PCE for enhanced electromechanical properties by doubling and halving the CNF and PCE amounts, as shown in **Supplementary Fig. 20**. Since CNF and PCE act as insulators for the electrode, their proportion needs to be minimized in comparison to MXene. Considering the spray-coating capability and the electrical conductivity of the bioelectrode, we determined the optimized mass ratio of MXene to be 82 wt% of the total electrode composition. Maintaining this MXene weight ratio (82 wt%), we precisely adjusted the amounts of CNF and PCE in the electrode to further improve its environmental stability. By comparing the environmental stability of bioelectrodes with different CNF and PCE ratios, we identified the weight ratio of 24:4:1 (MXene:CNF:PCE) as optimal for the FLEXER system.

Supplementary Fig. 20. MX-CNF-PCE electromechanical property test change ratio of CNF and PCE. Normalized resistance stability measurement against (a) tensile strain, (b) cyclic bending, (c) 100% humidity environment, and (d) a saline droplet.

4. According to the Bragg diffraction formula, there is a problem with the calculation of layer spacing in XRD, and the authors should give a wide range of XRD patterns. In addition, the authors need to prove their structure with detailed tests and add the reason for their increased interlayer spacing?

→ Most importantly, the XRD data depicted in **Figure 3i** illustrate that there is an alteration of the MXene interlayer distance within the (002) peak position, occurring in the θ range of 3 to 8 degrees. The determination of layer spacing in XRD should adhere to the formula $\lambda = 2d_{200} \sin \theta$, yielding interlayer spacings of 3.04, 3.10, and 3.78 nm for MXene, MX-CNF, and MX-CNF-PCE, respectively (*Sci. Adv.*, **8**, eab15299 (2022); *Nat. Commun.*, **13**, 4564 (2022); *Adv. Mater.*, **30**, 1707334 (2022); *Adv. Mater.*, **28**, 1517–1522 (2016)). To provide a more comprehensive understanding of the XRD data, we have included the complete spectrum of XRD data as illustrated in the revised figure. The shift of the distinctive (002) peak in the XRD pattern indicates the arrangement of MX-CNF-PCE interlayer gaps, confirming the structured layout of the MX-CNF and MX-PCE composite. This structural configuration arises from the strong bonding between each CNF and PCE component with the MXene in its dispersed form (Park et al., *Sci. Adv.* **8**, eab15299 (2022)).

Line 297-298: These shifts correspond to an increase in the interlayer spacing from 3.04 nm to 3.10 nm and 3.78 nm, respectively.^{22,45-57}

Supplementary Fig. 21. Extended XRD patterns of MX, MX-CNF, and MX-CNF-PCE.

Reply to Reviewer #2

The authors have developed a fabric based MXene electrode (FLEXER), which aims to improve comfort in traditional wearables. FLEXER offers excellent conductivity, low impedance, and shape-morphing capability, making it particularly suitable for dynamic users. The inclusion of pneumatic actuation as a novel mounting method for ECG and EMG biosensing is highly interesting. Therefore, I recommend this paper for publication in Nature Communications after necessary revisions.

→ We greatly appreciate the robust support provided by the reviewer in Nature Communications for our work. Based on their insightful feedback, we have meticulously revised the manuscript to further solidify our findings and conclusions.

1. The MX-CNF-PCE solution was applied to one leg of the FLEXER structure, leaving the other two legs uncoated. This asymmetrical coating may cause the center electrode to deviate from its intended perpendicular axis relative to the FLEXER's contact plane. Testing and an explanation of this issue are recommended.

→ Thank you for the feedback. The amount of MX-CNF-PCE applied to the FLEXER structure is negligible, 0.05 g, which constitutes approximately 2% of the total weight of FLEXER (**Figure R1**). Additionally, we analyzed top-view activation videos of both electrode-coated and non-coated FLEXER and evaluated the deviation of the center electrode (CE) from its center. As shown in **Supplementary Fig. 10**, we found that the deviation is negligible and less than the radius of the CE, indicating minimal impact on the positioning of the CE after activation with the electrode materials. To clarify this point, we have included **Supplementary Fig. 10** and the following statement in the main manuscript.

Figure R1. Mass change after spray-coating the MX-CNF-PCE on one side of SLs of FLEXER.

Supplementary Fig. 10. Comparison of the CE position before and after the activation of the FLEXER in cases of (a) without and (b) with an MX-CNF-PCE coating.

Line 171-178: Moreover, we evaluated the deviation of the CE from its center after the activation of FLEXER because MXene was coated on only one SL of the FLEXER, extending from the SL to the AF as a conductive

trace. **Supplementary Fig. 10** presents top-view images comparing the activation of MXene-coated and non-coated FLEXER samples. The CE diameter of the FLEXER is 10 mm, and the center of the CE remained aligned with the inner contact area after activation. This demonstrates the consistent movement of the FLEXER system along with the electrode materials.

2. On line 155, Page 5, the authors claimed that "The FLEXER also achieved a rapid actuation speed of ~20 cm/s by deploying the CE in < 0.2 s (Fig. 2b)". However, upon examining the data (both distance and angle) presented in Figure 2, it becomes evident that it would take at least 400 ms for the FLEXER to fully deploy the CE.

→ Thank you for your comments. We acknowledge the oversight in our manuscript, and it should be corrected to state: "The FLEXER also achieved a rapid actuation speed of approximately 12 cm/s by deploying the CE in less than 0.4 seconds (**Fig. 2b**, bottom left graph).

Line 158-159: The FLEXER also achieved a rapid actuation speed of ~12 cm/s by deploying the CE in <0.4 s (**Fig. 2b**, bottom left graph).

3. The author should conduct tests on the electrical response of their bioelectrodes using artificial sweat instead of a saline solution. This is because the ingredients of sweat and saline solution are significantly different and sweat can potentially impact the performance of the electrodes.

→ We acknowledge the significance of this comment and conducted an additional experiment using an artificial sweat formula referenced from the literature (*Toxicology in Vitro*, 24 (2010) 1790–1796). The composition of the artificial sweat included 22 mM urea, 5.5 mM lactic acid, 3 mM NH_4^+ , 0.4 mM Ca^{2+} , 50 μM Mg^{2+} , 25 μM uric acid, 100 μM glucose, 8 mM K^+ , 35 mM Na^+ , which consists of various electrolytes (NaCl, Na_2SO_4 , NaHCO_3 , KCl, NaCl, $\text{MgCl}_2 \cdot 6\text{H}_2\text{O}$, NaH_2PO_4 , CaCO_3 , and NH_4OH).

After submerging the three different electrodes (MX, MX-CNF, MX-CNF-PCE) in the artificial sweat for intervals of 0, 4, 8, 12, 16, and 20 minutes, we measured the change in electrical resistance. The results showed that the change in resistance of the three electrodes was larger than what we previously observed with a saline solution. However, the resistive environment characteristics of the MX-CNF and MX-CNF-PCE electrodes, compared to MX, remained consistent. Consequently, this experiment demonstrated that the MX-CNF-PCE electrode could effectively function as a bio-electrode for FLEXER. To clarify this, we have added the data in **Figure 3h** and edited the manuscript accordingly.

Line 276-278: To assess the stability of MX-CNF-PCE in response to perspiration, its electrical characteristics were observed in artificial sweat.⁴⁴ After wetting the three MXene-based electrodes with artificial sweat (**Supplementary Fig. 19**), we monitored their electrical response (R/R_0) for 20 minutes (**Fig. 3h**).

Fig. 3 | Chemical analysis of the MX-CNF-PCE electrode. Normalized resistance stability measurement under four different conditions: (h) artificial sweat droplet.

4. The authors showcase FLEXER's effectiveness in wearable biosensing during motion, but concerns remain about its performance during large movements and compatibility with looser clothing. Further explanation is recommended.

→ We appreciate your insightful input and viewpoint. The physiological signal remained notably distinct and resilient against disruption during three common daily activities: standing, walking, and driving, as demonstrated with our loose-fitting t-shirt (Fig. 4f). To accommodate even looser fits in future cases, FLEXER could modify a design factor, such as the diameter of the anchoring frame (AF), to enhance contact distance. Furthermore, for more vigorous movements, we could incorporate accelerometers and identify the optimal timing to promptly activate FLEXER for real-time physiological signal measurement.

Fig. 1 | Real-time electrophysiological monitoring of ECG using FLEXER. (f) Real-time measurement of ECG using multi-FLEXER in situations involving standing, walking, and driving, both in the inactive and active status of FLEXER.

5. Only 16 samples are displayed in Figure 2e. It is suggested to remove the three sets of ellipses.

→ Thank you for your feedback. We have removed the three sets of ellipses as per your suggestion.

Fig. 2 | Physical and design analysis of FLEXER. (e) Evaluation of 16 multi-legged symmetric FLEXER variants. (f) Multi-legged symmetric FLEXER contact force based on an applied pneumatic pressure of 5 psi. The plane represents equation (1).

6. The top row pictures in Figure 2a lack clarity. It is advised to avoid using a black background.

→ We appreciate your feedback. The decision to use a black background for the white pneumatic actuator was informed by a methodological approach that is widely adopted, as evidenced in various references (*Adv. Funct. Mater.*, **24**, 2163–2170 (2014), *Nat. Commun.*, **9**, 878, (2018)). In response to your suggestions, we have provided further clarifications on the visual representation and enhanced the readability of **Figure 2a** by adjusting its contrast. Thank you once again for your valuable input.

Fig. 2 | Physical and design analysis of FLEXER. Kinetic analysis of pneumatic actuation: Three actuation snapshots of (a) the straightened simple linear shape of FLEXER.

REVIEWERS' COMMENTS

Reviewer #1 (Remarks to the Author):

Authors have satisfactorily answered all the queries and modified the manuscript as per the suggestions. The manuscript may be now accepted for publication.

Reviewer #2 (Remarks to the Author):

The author provided excellent responses, but I still have a concern about question 4. Given that FLEXER is a novel mounting method for biosensing, its effectiveness in wearable biosensing during motion is crucial. The authors've examined various leg widths and quantities in FLEXER, yet they did not discuss leg length. I am curious if longer legs could extend the biosensor's range, making it suitable for looser clothing or more vigorous movements.

REVIEWERS' COMMENTS

Reply to Reviewer #1

Reviewer #1 (Remarks to the Author):

Authors have satisfactorily answered all the queries and modified the manuscript as per the suggestions. The manuscript may be now accepted for publication.

→ We appreciate the comments from the reviewer.

Reply to Reviewer #2

Reviewer #2 (Remarks to the Author):

The author provided excellent responses, but I still have a concern about question 4. Given that FLEXER is a novel mounting method for biosensing, its effectiveness in wearable biosensing during motion is crucial. The authors've examined various leg widths and quantities in FLEXER, yet they did not discuss leg length. I am curious if longer legs could extend the biosensor's range, making it suitable for looser clothing or more vigorous movements.

→ The semicircular legs (SLs) maintain their geometric structure, hence their length remains fixed within the diameter of the anchoring frame (AF) (**Supplementary Fig. 8**). This is illustrated by the relationship between the AF diameter and contact distance, as well as the correlation between the length of SLs and contact distance, demonstrated in **Supplementary Figure 12**. The slope of the graph is 0.3, which could be shown as following equation.

$$\text{Slope} = \frac{\Delta y}{\Delta x} = \frac{0.3 \text{ cm}}{1 \text{ cm}} = +30 \%$$

As the 1 cm of AF increase, the contact distance will increase in 30 % of it which dedicates the longer contact of FLEXER for looser fit clothes.

Supplementary Fig. 8. AF, SLs, and CE of FLEXER

Supplementary Fig. 12. Maximum contact distances for different FLEXER sizes. The ratio of the AF diameter to the CE diameter of FLEXER is 6.8, and the ratio of the CE diameter to the SL width is 2.

Line 198-200: Additionally, the maximum contact distance of the FLEXER electrode depends on the FLEXER size, as shown in **Supplementary Fig. 12**. From a geometric perspective, the contact distance of FLEXER is ultimately linked to the AF diameter because other factors, such as the length of SLs, are all constrained within the boundaries of the AF diameter. When the AF diameter of the FLEXER varied from 8 to 13 cm, the contact distance increased from 1.7 to 3.2 cm, all at a constant pressure of 10 psi.